# Research Progress on Laser-Assisted Precision Machining Technology

**DOI:** 10.3390/mi16020173

**Published:** 2025-01-31

**Authors:** Qimeng Liu, Jie Liu, Zhe Ming, Bo Cui, Jian Wang

**Affiliations:** 1School of Mechanical and Civil Engineering, Jilin Agricultural Science and Technology University, Jilin 132101, China; jlnkjxgc@126.com (Z.M.); db0425@126.com (B.C.); 17743023393@163.com (J.W.); 2Advanced Manufacturing Technology Engineering Research Center for Key Components of Agricultural Machinery Equipment, Jilin Agricultural Science and Technology University, Jilin 132101, China; 3Information Management Center, Jilin Agricultural Science and Technology University, Jilin 132101, China

**Keywords:** laser-assisted cutting, laser-assisted grinding, laser-assisted milling, laser-assisted drilling

## Abstract

As a revolutionary advanced manufacturing technology, the core of laser-assisted machining technology lies in the innovation of traditional machining technology. It uses a laser to precisely modify the surface of the workpiece material through clever integration. This process can lead to significant changes in the microstructure, thermodynamic properties, and physical properties of the workpiece’s cutting layer, greatly facilitating the removal of material by the cutting tool. Especially for difficult-to-process materials, this technology can effectively solve the processing problems they face. This not only improves the processing accuracy, but also ensures the processing quality, making it possible to process difficult-to-process materials with high precision and high quality. This article comprehensively summarizes the latest research progress in laser-assisted precision machining technology and deeply summarizes the specific mechanism of laser-assisted machining technology on the surface of workpiece materials. The influence of laser-assisted machining technology on the cutting characteristics of machining materials is elaborated in detail, providing a theoretical basis for the application of this technology. Finally, the future development direction of laser-assisted machining technology is prospected.

## 1. Introduction

With the booming development of high-end equipment manufacturing industries such as aerospace, the requirements for the performance and reliability of their equipment components are becoming increasingly stringent. Therefore, a series of difficult-to-machine materials with excellent properties such as high strength and hardness have emerged, which undoubtedly bring unprecedented difficulties and challenges to the field of cutting and machining. How to effectively improve the machining accuracy, quality, and efficiency of difficult-to-machine material parts has become a key issue that urgently needs to be addressed.

When discussing the classification and processing difficulties of difficult-to-machine materials, we can roughly divide these materials into five categories: high-temperature alloys, titanium alloys, stainless steel, high-strength steel, and hard and brittle materials. The reason that these materials are difficult to process is mainly due to their common characteristics, such as high material strength, high hardness, insufficient thermal conductivity, severe hardening during processing, and high chemical activity. These characteristics work together to cause serious cutting deformation, a significant increase in cutting force, a sharp rise in cutting temperature, and accelerated tool wear of parts during the machining process.

In the precision machining process of parts, we can vividly compare cutting tools and workpiece materials to “spears” and “shields”. Between these two, the processing technology is like a sturdy bridge, connecting and coordinating their relationship. How to effectively enhance the sharpness of the “spear” or cutting tool, while cleverly weakening the resistance of the “shield” or workpiece material, has become the core of improving the efficiency, accuracy, and quality of part processing. This is clearly and intuitively demonstrated in Figure 1.

In the manufacturing industry, improving the cutting performance of cutting tools is a key strategy for optimizing the machinability of difficult-to-machine materials. Superhard cutting tool materials, such as Polycrystalline Cubic Boron Nitride (PCBN), Polycrystalline Diamond (PCD), and ceramic cutting tools, significantly enhance the cutting efficiency of cutting tools due to their excellent hardness and wear resistance. In addition, surface coating or surface modification treatment of cutting tools, such as using titanium nitride (TiN) coating, micro-textured cutting tools, or a non-metallic compound surface layer, can not only enhance tool hardness, but also reduce friction and wear and lower the cutting temperature during the cutting process, representing the latest technological development to improve cutting performance. On the basis of existing cutting tools, by carefully designing and optimizing the geometric structure of a cutting tool, such as designing cooling channels inside the cutting tool, the cutting temperature can be effectively reduced, and the service life of the tool can be extended. This is also an effective method to improve cutting performance. The application of these technologies provides multidimensional solutions to improve the performance of cutting tools, further promoting technological innovation and development in the manufacturing industry [1,2,3,4,5,6,7,8,9,10,11,12].

In exploring strategies to enhance the machinability of difficult-to-machine materials, besides improving cutting tool materials, surface coatings, surface modification treatments, and cutting tool structures, implementing modification treatment techniques on the surface of difficult-to-machine materials is also an effective approach. By chemical or physical means, the microstructure, chemical composition, and thermodynamic and physical properties of the cutting layer of difficult-to-machine materials can be changed, making the material of the layer to be processed easier to remove. This surface modification and weakening treatment technology has shown great potential for application in the field of cutting difficult-to-machine materials. For example, advanced modification techniques such as high-energy fields (such as lasers) can effectively weaken the microstructure and mechanical properties of the cut surface, thereby significantly improving the machinability of difficult-to-machine materials [13,14,15,16,17].

This article comprehensively reviews the cutting-edge research progress of laser-assisted precision machining technology and deeply summarizes the complex mechanism of laser modification treatment technology on the machining surface of workpiece materials. The article elaborates in detail on how laser-assisted machining technology significantly affects the cutting characteristics of processed materials and reveals its influencing laws. At the end of the article, this paper provides an in-depth outlook on the future development of laser-assisted machining technology, aiming to provide valuable reference and inspiration for researchers in this field.

## 2. Laser-Assisted Machining

Laser-assisted machining (LAM), with its innovative manufacturing philosophy, has led the trend in manufacturing processes. Its core advantage lies in accurately injecting laser energy into the processing area, preheating and softening materials that are difficult to process in advance, and optimizing their characteristics during deformation. This advanced technology enables materials to be efficiently removed in a plastic manner, significantly improving their processing performance under temperature induction and achieving a qualitative leap in material processing. With the continuous advancement of technology, laser-assisted machining has gradually become an indispensable part of modern manufacturing. It not only covers various composite machining technologies such as laser-assisted cutting, grinding, milling, and drilling, but has also won widespread favor in the industry with its excellent high precision, high efficiency, and unparalleled flexibility. This technology can meet the processing challenges of various materials, such as high-temperature alloys, titanium alloys, stainless steel, high-strength steel, and hard and brittle materials, all of which can be easily handled. Laser-assisted processing technology has been widely applied in various high-tech fields such as aerospace, automotive manufacturing, and medical equipment, demonstrating its strong vitality and broad application prospects [18,19,20,21,22,23,24,25,26,27].

### 2.1. Laser-Assisted Cutting

Laser-assisted cutting (LAC) is a cutting-edge manufacturing technology that combines laser technology with classic cutting processes. The core principle lies in the precise local preheating treatment of materials using high-energy laser beams, which increases the temperature of the material in the cutting area, thereby achieving lower hardness and increasing brittleness. After the material is preheated, the cutting tool can process the material more efficiently. Due to this change in material properties, the required cutting force and wear rate during the cutting process are significantly reduced. In addition, laser-assisted cutting technology can effectively reduce the heat-affected zone in the cutting area, further improving machining accuracy and surface smoothness. This technology is particularly suitable for materials that are difficult to process, such as titanium alloys, nickel based alloys, etc., and can play an important role in improving cutting efficiency and processing quality.

In traditional laser-assisted cutting technology, the laser source is directly aimed at the surface of the workpiece and emitted, utilizing the preheating effect of the laser to reduce the surface hardness of the material and achieve the ultra-precision machining of difficult-to-machine materials. This laser-assisted cutting device usually consists of a machine tool and a laser module. The laser beam is irradiated onto the area to be cut, and the workpiece is softened by heating to a specific temperature. Then, the machine tool completes the cutting process, as shown in Figure 2a [28]. Traditional LAC technology changes the cutting performance of materials by directly irradiating the surface of workpieces with lasers, thereby improving the surface quality and machining accuracy of workpieces to a certain extent. However, it also has some shortcomings: firstly, the distance between the laser spot and the tool is relatively large, reaching centimeter level. In order to achieve a softening effect on the workpiece, a large laser power is required, resulting in high energy consumption. Secondly, cutting fluid plays a difficult role in the cutting process because it may interfere with the laser path and increase the uncertainty of the laser focus position. Finally, due to a certain gap between the laser spot and the tool, machining the center position of the workpiece becomes impossible. In response to these issues, scholars have proposed in situ LAC technology, which allows the laser beam to penetrate the tool body and focus on the cutting position. Through the coupling effect between the laser and the tool, in situ heating of the workpiece material is achieved, making the temperature in the processing area precise and controllable, cleverly solving the technical challenges faced by traditional LAC. This processing principle is shown in Figure 2b [29].

Currently, numerous researchers are dedicated to applying laser-assisted cutting (LAC) technology to traditional materials with high processing difficulty, such as silicon nitride, ZrO_2_ ceramics, and fused quartz glass. The thermal effect of a laser is used to pretreat materials, locally soften them, effectively reduce their yield strength, and enhance their ductility. This innovative method significantly improves the surface quality of difficult-to-machine materials, providing new possibilities for high-precision machining. Song et al. [29] conducted in-depth research on the significant effects of conventional cutting techniques and classical laser-assisted cutting (LAC) on the surface quality of fused quartz machining. The experimental results revealed an important phenomenon: compared with traditional cutting methods, the surface cracks and grooves of fused quartz workpieces processed using classical LAC technology were significantly reduced, thereby significantly improving surface quality. This discovery effectively enhanced the performance of fused quartz glass during cutting, providing a valuable reference for improving the quality and efficiency of fused quartz processing technology. In exploring the field of composite material processing, Wang [30] conducted an in-depth analysis of the inherent defects of traditional milling and laser processes in processing composite materials. He ingeniously integrated laser and micro milling technology, conducting a detailed study specifically on thermosetting glass fiber reinforced plastic (GFRP) and carbon fiber reinforced plastic (CFRP) materials. In the study, Wang measured the morphology and size of the heat-affected areas of GFRP and CFRP during laser-assisted micro milling and explored the degree of influence of laser processing on the substrate materials in different areas. In addition, he systematically analyzed various aspects of the processing process and evaluated the feasibility of the process, providing valuable insights for the advancement of composite material processing technology. Liu et al. [31] used molecular dynamics simulation technology to investigate in depth the cutting mechanism of single crystal silicon under a high-temperature environment and the evolution trend of subsurface damage. Through conducting in situ laser-assisted cutting experiments, they found that as the cutting temperature increased, the shear resistance of the amorphous layer significantly decreased, and the self-lubricating effect of the amorphous layer became more significant, which had a positive effect on suppressing subsurface damage. Geng et al. [32] utilized the solid foundation of finite element theory to construct an accurate three-dimensional transient heat transfer model, aiming to investigate the intrinsic relationship of temperature response under various processing conditions. They not only explored the mechanism of material removal during in situ laser-assisted processing but also evaluated the possibility of achieving efficient processing. Through detailed characterization analysis of the processed surface, they found that under the influence of in situ laser-assisted processing, the dislocation movement rate in single-crystal silicon was significantly increased, which effectively promoted the plastic deformation of single crystal silicon. Further, X-ray diffraction pattern analysis revealed an important phenomenon: under the action of in situ laser, the residual stress on the surface of the workpiece was significantly reduced. Based on the results of processing experiments, they noticed that increasing the rotational speed has a positive effect on the plastic processing of single-crystal silicon workpieces. This not only suppresses the formation of surface micro pits but also reduces surface roughness and amorphous layer thickness with increasing rotational speed. These findings provide valuable guidance for optimizing processing parameters and improving processing quality. He et al. [33] conducted an in-depth study on the temperature field distribution during in situ laser-assisted ultra-precision cutting using a combination of finite element simulation (FEM) and experimental testing. They employed Smooth Particle Dynamics (SPH) technology to conduct a precise simulation analysis of the in situ laser-assisted cutting process of single-crystal silicon under different temperature conditions. Through experimental verification, they found that an in situ laser can effectively enhance the brittle ductile transition depth of single-crystal silicon to 378.11 nm, thereby achieving the plastic domain processing of single-crystal silicon. This discovery provides an important theoretical basis and technical support for the field of ultra-precision machining. Chang et al. [34] observed in their study on laser-assisted planing of alumina ceramics that compared to traditional planing methods, the axial force was significantly reduced by 20%, the radial force was reduced by 22%, and the surface roughness Ra was reduced by more than 50%. These improvements not only significantly improve the smoothness of the surface, but also greatly enhance the overall surface integrity of the workpiece. In Tian et al.’s study [35], they conducted experiments on heating-assisted laser polishing of AISI4140 steel and MP35N nickel-based alloy. The experimental results show that compared with traditional polishing processes, heating-assisted laser polishing significantly reduces tool wear and improves the integrity of the machined surface. However, this process resulted in a certain degree of increase in surface residual stress. In response to the challenges of high hardness, difficult dressing, and low dressing efficiency of superhard abrasive grinding wheels, Zhang et al. [36] conducted experimental research on the heating-assisted laser turning dressing of metal bond cubic boron nitride (CBN) grinding wheels. The research results indicate that while ensuring the quality of the dressing, compared to traditional diamond tool dressing methods, heating-assisted laser turning dressing technology can not only significantly reduce dressing time and improve dressing efficiency but also effectively extend the service life of dressing tools. This discovery provides new ideas for the efficient dressing of superhard abrasive grinding wheels and has important industrial application value. A comparison of tool wear, machining quality, and cutting force data between conventional cutting and laser-assisted cutting is shown in Figure 3.

The main advantage of laser-assisted cutting technology is it significantly improves the surface quality of the processed material. The use of high-temperature preheating and softening materials generated by lasers can effectively reduce cutting forces and tool wear while improving machining efficiency. In addition, laser-assisted cutting technology can effectively reduce the heat-affected area during the machining process and maintain the original properties of the material, which is crucial for high-quality machining. In terms of application fields, laser-assisted cutting technology has been widely used in various industries such as aerospace, automotive manufacturing, and mold manufacturing. It has demonstrated its unique advantages in processing difficult-to-machine materials such as titanium alloys and stainless steel, providing innovative solutions for the manufacturing of complex structural components. The development trend of laser-assisted cutting technology is expected to focus on increasing laser power, optimizing process parameters, and expanding application scope. With the continuous advancement of laser technology, higher-power lasers will be developed to meet a wider range of processing needs. Meanwhile, by precisely controlling the laser power and cutting parameters, the processing efficiency and quality can be further improved. In addition, the application prospects of laser-assisted cutting technology in micro/nano-processing, biomaterial processing, and other fields are also promising, which is expected to bring revolutionary changes to more industries.

### 2.2. Laser-Assisted Grinding

Laser-assisted grinding (LAG) is an innovative manufacturing technology that combines laser technology and traditional grinding processes. The core of this technology is to first preheat the material surface through a high-energy laser beam, aiming to reduce the hardness of the material and remove surface defects. Then, fine machining is carried out using grinding tools; the machining principle is shown in Figure 4. The laser beam acts on the surface of the material at high temperature, rapidly heating up to a softened or evaporated state, effectively reducing friction and wear during the grinding process, thereby improving processing efficiency. The accurate control of key parameters such as laser beam power, focusing position, and movement speed is crucial in the laser-assisted grinding process to ensure the reliability of machining quality. The preheating effect of the laser beam significantly reduces the contact stress between the grinding tool and the material, lowers the grinding force, reduces the accumulation of heat, and effectively avoids the risk of material surface burns and deformation. In addition, laser-assisted grinding technology can improve the material removal rate and shorten the processing cycle, thereby enhancing processing accuracy and surface quality, bringing revolutionary progress to the field of high-precision manufacturing.

Laser assisted grinding (LAG) technology was first proposed by Tian and Shin [37]. Before grinding ceramic materials, laser preheating is used to raise the local temperature of the grinding area to above 1600 °C. This process can promote the workpiece to transition from a brittle state to a plastic state during the grinding process, thereby significantly improving the surface quality of material processing, reducing the wear of grinding tools, and effectively reducing the cost of precision machining of difficult-to-machine materials. Liu Wei [38] used laser-assisted grinding to process alumina ceramics, which can change the material removal method and transform the brittle removal of ceramics into plastic removal. Compared with conventional grinding processing, the brittle fracture of the machined surface morphology is reduced, the surface roughness value is lower, the surface quality is better, and the service life of the grinding wheel is extended. Researchers such as Ma Zhelen [39] constructed a LAG processing device aimed at improving the problems of difficult processing and poor surface quality of alumina ceramics. The experimental results show that compared with conventional grinding techniques, the brittle fracture phenomenon in the machined surface morphology using LAG technology is significantly reduced, the surface roughness Ra value is reduced, the surface quality is significantly improved, and the service life of the grinding wheel is correspondingly extended. Xiao Guijian [40] studied silicon carbide ceramics and used laser-modified grinding technology to modify them through laser irradiation, and then conducted grinding experiments on silicon carbide ceramics. Compared with ordinary grinding, the grinding force, surface roughness, surface morphology, and subsurface damage of silicon carbide ceramic specimens were studied. The experimental results show that compared with ordinary grinding, laser-modified grinding can effectively reduce the normal grinding force, tangential grinding force, and surface roughness. Ma et al. [41] conducted a comparative study on the surface quality of alumina ceramics between LAG and traditional grinding techniques. Research has found that the surfaces of traditional ground alumina ceramics often exhibit brittle fracture characteristics, with sharp pit edges and a tearing and dissociation state in the cross-section. After adopting LAG technology, the surface presents a molten state, and compared with traditional grinding, the brittle fracture phenomenon on the surface is greatly reduced. The surface roughness Ra value and sub-surface damage depth are significantly reduced. He Yi [42] conducted experiments on the laser-assisted CBN sand belt grinding of TC4 titanium alloy and studied the influence of laser power on grinding force, material removal behavior, and surface integrity. The results indicate that increasing laser power can improve the degree of material softening modification, reduce the grinding force and its fluctuation amplitude. Xu et al. [43] introduced laser preheating technology before grinding to achieve the efficient and low-cost precision machining of zirconia ceramics. The research results confirm that the process of grinding after laser preheating can effectively reduce grinding force and energy, and this process can significantly overcome the limitation of the high specific grinding energy of difficult-to-machine materials such as ceramics in the grinding process. ZHANG et al. [44,45,46] performed laser-assisted grinding on silicon nitride ceramics and aluminum oxide ceramics. They used laser irradiation to create thermally induced cracks on the surfaces of the two ceramics and then ground the ceramics. They found that laser-assisted grinding can reduce the grinding force and surface roughness. GUERRINI et al. [47] conducted experimental research on silicon nitride ceramics, first using a laser heat source to process the surface of silicon nitride ceramics to generate thermally induced cracks and then grinding them. The research results indicate that when the laser power is 2 kW, a significant oxide layer appears on the surface of the workpiece; when the power reaches 3 kW, obvious cracks appear on the surface of the workpiece; meanwhile, laser-assisted grinding processing effectively reduces the maximum and average grinding forces, with a maximum reduction of 26% to 27%. Similarly, Li et al. [48] used laser ablation on the surface of silicon nitride ceramics and processed different microstructures on it. The subsequent grinding test results showed that the microstructure of the silicon nitride ceramic surface significantly reduced the grinding force, with a maximum reduction of 63%. Another way is to build a composite processing system of laser and grinding, where laser processing and grinding are carried out simultaneously online, which can be called online laser-assisted grinding. Li Zhipeng et al. [49,50] conducted laser-assisted grinding on reaction-sintered silicon carbide ceramics. Compared with the silicon carbide base material, laser irradiation caused a modified layer on the surface of the silicon carbide, which reduced its hardness and increased its toughness. This indicates that laser-assisted grinding was very effective in reducing the surface roughness and subsurface damage of the workpiece, significantly improving its surface processing quality. MA et al. [51] developed a novel laser-assisted grinding system that combines laser and CBN grinding wheels. Through local laser heating and rapid removal of the grinding wheel, the laser-assisted grinding of ceramics was achieved, and the laser-assisted grinding and conventional grinding of zirconia ceramics were carried out. The experimental results show that compared with ordinary grinding, laser-assisted grinding transforms zirconia ceramics from brittle fracture to plastic fracture, reduces material subsurface damage, achieves plastic removal, and improves the surface integrity of zirconia ceramics. RAO et al. [52] conducted laser-assisted grinding on reaction-sintered silicon carbide ceramics and studied the wear of diamond grinding wheels at different laser processing temperatures. Experiments have shown that at room temperature, the abrasive grains on the surface of the grinding wheel exhibit fracture and flattening phenomena. During laser-assisted grinding, silicon carbide ceramics soften and adhere to the grinding tool at high temperatures, resulting in abrasive grain bonding. As the laser temperature increases, the detachment and wear of diamond abrasive particles decrease, significantly extending the service life of the grinding wheel. Ma Zhelen et al. [44] conducted comparative experiments between laser-assisted grinding and conventional grinding on alumina ceramics. Research has shown that in laser-assisted grinding, the proportion of plastic removal of alumina ceramics is significantly increased, brittle fracture on the material surface is reduced, the surface roughness value of the material is lowered, and the service life of the grinding wheel is extended. A comparison of machining quality data between conventional grinding and laser-assisted grinding is shown in Figure 5.

Through analysis of the current international research situation, it can be concluded that laser-assisted grinding can significantly reduce the surface roughness of hard and brittle materials, improve processing quality, reduce grinding force, and decrease tool wear, thereby greatly improving the service life of tools and reducing processing costs. This is conducive to expanding the industrial application scope and development prospects of hard and brittle materials. However, there are still certain issues that require further research work: (a) Due to the characteristics of laser processing, the temperature of the laser beam is high and difficult to control, which can lead to excessive and uneven modification of hard and brittle materials. It is necessary to study how to accurately control the temperature to achieve uniform modification effect, in order to reduce the difficulty of subsequent material removal. (b) In laser-assisted grinding, the process parameters include laser processing parameters and grinding processing parameters. The material removal mechanism is affected by multiple factors coupled and interfered, and the precise evaluation model of the critical cutting depth for brittle plastic transition is not yet clear. It is necessary to find the correlation between the thickness of the modified layer of hard and brittle materials and the grinding depth, in order to achieve higher processing quality and efficiency. (c) The composite device of laser systems and grinding systems is difficult to build, the device is complex, and its safety is difficult to guarantee. It is necessary to strengthen the production research of laser-assisted grinding composite devices, enhance the stability, safety and accuracy of the composite devices, and expand the application scope of laser-assisted grinding technology.

### 2.3. Laser-Assisted Milling

Laser-assisted milling (LAMill) is an advanced manufacturing process that combines traditional milling techniques with laser technology. This technology significantly improves the material removal efficiency, reduces the heat-affected area during the machining process, and minimizes tool wear by introducing laser heating during the milling process, thus demonstrating unique advantages in the field of hard material machining [53,54,55].

The principle of laser-assisted milling technology is rooted in the mystery of the interaction between the laser and materials. In this precise machining process, a laser beam is first focused on the surface of the workpiece, causing a sharp increase in the temperature of the material in that area until it reaches the melting or boiling point, thus forming a molten or vaporized layer. Subsequently, the milling tool gently cuts into the workpiece with a lower cutting force, cleverly removing materials that have already melted or evaporated. Thanks to the preheating effect of the laser, the hardness and strength of the material are significantly reduced, making the milling process smoother. The machining principle is vividly demonstrated in Figure 6. Laser-assisted milling technology, with its synergistic effect of laser preheating and milling, has achieved high efficiency, low wear, and high-quality machining results, which can be regarded as a revolutionary breakthrough in the field of hard material processing. With the continuous advancement of laser technology and CNC machining technology, laser-assisted milling technology (LAMill) is expected to be widely applied in key fields such as aerospace, automotive manufacturing, and energy, injecting new vitality into the development of industrial manufacturing.

Kumar and Melkote [56] delved into the laser-assisted milling (LAMM) of A2 tool steel in their meticulous research and achieved remarkable results. The experimental data clearly and powerfully reveal that compared with traditional milling processes, LAMM technology exhibits significant advantages in reducing cutting forces, with an astonishing average reduction of 69%. At the same time, the material removal rate achieved an astonishing six-fold increase. It is worth mentioning that the wear level of LAMM cutting tools is significantly lower than that of traditional milling tools, which is particularly noteworthy. Woo and Lee [57] designed an innovative laser-assisted machining machine (LAMM) device with a controllable laser heat source, specifically designed for the processing of high-strength alloy Inconel 718. Their research results show that when the milling position angles are set to 75°, 45°, and 15°, compared with traditional milling processes, the LAMM process improves surface quality by 7%, 40%, and 24% during the machining process, respectively. This significant discovery opens up a new perspective for improving processing quality and provides a valuable reference for research and practice in related fields. Cha et al. [58] proposed an innovative preheating method based on laser power, which can effectively transfer thermal energy to the surface of the workpiece. They successfully validated the effectiveness of this method during the processing of Si_3_N_4_ workpieces. In addition to this, researchers also delved into the influence of different process parameters on the surface roughness and cutting force of workpieces. By using the Taguchi method and orthogonal analysis, they provided an optimal solution for optimizing the laser-assisted grinding (LAMM) process parameters of AISI 1045 steel. In their in-depth research, Bermingham et al. [59] particularly focused on the application of laser-assisted milling technology in the processing of martensitic precipitation hardening stainless steel. They carefully analyzed the specific impact of laser preheating on the tool wear mechanism and compared it in detail with the results of milling experiments conducted at standard room temperature conditions. Research has found that laser-assisted milling technology significantly reduces cutting forces by up to 33% and effectively suppresses high-frequency chatter phenomena that often occur during room temperature cutting, significantly extending the tool’s service life. These findings not only highlight the significant potential of laser-assisted milling technology in improving machining efficiency, reducing tool wear, and enhancing machining quality, but also provide valuable experimental data and a theoretical basis for future industrial applications. A comparison of tool wear and machining quality data between conventional milling and laser-assisted milling is shown in Figure 7.

Laser-assisted milling technology, as an innovative technology in modern manufacturing, has attracted considerable attention for its development history and application prospects. This section provides an overview of the key research achievements in laser-assisted milling technology and delves into its unique advantages and potential challenges. Through a systematic analysis of existing literature, this article proposes the significant contribution of laser-assisted milling technology in improving machining efficiency, enhancing surface quality, and expanding material processing range. Overall, laser-assisted milling technology demonstrates outstanding potential in high-precision and complex shape machining, providing new ideas and solutions for the future precision machining field. With the continuous advancement and optimization of technology, it is expected that laser-assisted milling will play a more important role in high-end manufacturing fields such as aerospace, automotive manufacturing, and medical devices.

### 2.4. Laser-Assisted Drilling

Laser-assisted drilling (LAD) technology is an advanced manufacturing technology that combines laser and mechanical processing. It uses a high-power laser beam to focus on the surface of the material, instantly reaching the melting point in a local area, forming a molten material, and then discharging the molten material out of the hole through a high-speed rotating drill bit, achieving high-quality drilling processing. This technology is particularly suitable for difficult-to-machine materials such as titanium alloys, stainless steel, etc., which can improve processing efficiency while maintaining material integrity. The principle of laser-assisted drilling can be divided into the following steps: (a) The laser beam is guided to the surface of the target material, and the energy is concentrated in a very small area through a focusing system. (b) The material rapidly melts or evaporates under high energy density, forming small pores. (c) The mechanical drill bit synchronously enters the melting zone and discharges the melted material; (d) As the drill bit goes deeper, the laser beam continues to act on the newly formed hole’s wall until the drilling is completed, and this processing principle is shown in Figure 8. The advantage of this technology is that it can reduce machining stress and material hardening, improve the surface quality of the hole wall, reduce tool wear, and is suitable for high-precision, deep hole machining. Meanwhile, laser-assisted drilling can effectively control aperture tolerances and is suitable for precision manufacturing needs in fields such as aviation and aerospace. Due to the difficulty of laser transmission to the material to be removed in the drilling area, there is currently not much research on LAD.

Zheng et al. [60] conducted experimental research on drilling key automotive components using laser heating-assisted drilling technology. Compared with conventional drilling, 40Cr, 45 steel, and stainless steel increased the drilling aperture by 50.5%, 52.2%, and 51.4%, respectively. In terms of drilling efficiency, QT600, 45 steel, and stainless steel increased by 19.3%, 16.3%, and 39.9%, respectively. Similarly, Zhang et al. [61] studied 41Cr4 and C45E4. Experimental research has been conducted on LAD technology using stainless steel as the workpiece material. Compared with conventional drilling, 41Cr4, and C45E4, the drilling time of stainless steel was reduced by 18.6%, 16.3%, and 39.9%, respectively, indicating that laser can significantly improve the efficiency of drilling difficult-to-machine materials. Jen et al. [62] applied a circular laser spot to improve drilling performance, obtained the influence law of laser power and laser spot size on heating temperature, and improved drilling quality and efficiency. Due to the fact that a laser can only heat and soften the surface layer of the workpiece material in LAD, and as drilling progresses, a laser cannot heat the material inside the hole, the further improvement of LAD processing efficiency is limited. Mohammadi and Patten [63] from Western Michigan University proposed an in situ laser-assisted drilling technique that uses a focused laser beam guided through an optically transparent diamond tool to soften the workpiece material and process the workpiece in a non-ablative state. The extremely high compressive and shear stresses generated by the cutting edge are beneficial for improving the machinability and laser absorption rate of the workpiece, thereby achieving high-quality precision holes and delamination-free blind holes in carbon fiber composite materials. Studying the quality of hole machining under different laser powers, it was found that without laser action, the roundness of the drilled hole was poor. After using laser, the roundness of the drilled hole was better. Although there were still cracks at the edges, the edge quality gradually improved with the increase in power. A comparison of machining quality data between conventional drilling and laser-assisted drilling is shown in Figure 9.

In the field of laser-assisted drilling, laser technology mainly achieves an initial stage of efficient drilling by rapidly heating and softening the surface material of the workpiece. However, as the drilling depth gradually increases, the influence of laser heating on the material inside the hole gradually weakens, and it cannot further promote an improvement in processing efficiency. Currently, there is a relative lack of research reports on drilling force, tool wear, hole roundness, and surface roughness in laser heating-assisted drilling processing. Further deepening and expansion of the research in these fields is urgently needed.

### 2.5. Other Laser-Assisted Processing Methods

Laser-assisted processing technology is also applied to other processing methods, such as polishing. Laser-assisted polishing (LAP) can increase the plastic deformation of the surface layer of a workpiece material, significantly reduce tool wear compared to traditional polishing, improve the integrity of the machined surface, and effectively reduce the surface roughness of the workpiece. Tian and Shin [35] conducted experimental comparisons between AISI 4140 traditional polishing and LAP. As the laser power increased, the roughness of the workpiece gradually decreased. When the laser power was 500 W, compared with traditional polishing (0 W), the surface roughness Ra decreased by about 35%. Barnes et al. [40], in polishing aluminum/silicon carbide composites, found that laser pretreatment of the layer to be processed effectively reduced the surface hardness of the workpiece. Under the selected preheating conditions, with the reduction in the feed force, the wear of the rear tool face decreased by nearly 50%. A comparison of quality data between conventional polishing and laser-assisted polishing is shown in Figure 10.

Laser-assisted machining technology has been widely used in manufacturing due to its advantages such as high precision, high efficiency, non-contact processing, and wide applicability. However, it also has limitations such as high equipment costs and high energy consumption, requiring the selection of appropriate technical solutions based on specific application scenarios. Table 1 describes the laser specifications (such as wavelength, power, and pulse parameters) and the advantages of laser-assisted processing technology.

## 3. Summary and Prospect

Laser-assisted machining (LAM) technology, as an innovative precision machining method, has attracted widespread attention in the field of high-precision machining of difficult-to-machine materials due to its significant advantages in reducing tool wear and improving workpiece surface quality. This technology adds laser modules to traditional cutting, grinding, milling, and drilling equipment, which not only simplifies and speeds up the installation process but also effectively addresses the challenges faced by some difficult-to-machine materials in ultra-precision machining. Although the benefits brought by LAM technology are obvious, most of the current academic research on it is still in the experimental stage to verify its superior performance. In terms of the in-depth exploration of machining process details, tool wear mechanisms, and other aspects, existing research appears insufficient. The existing research framework has not fully covered the complexity and diversity of LAM technology, especially in the mechanism analysis of its processing, and there is still a lot of exploration and development space. In order to gain a deeper understanding of the physical phenomena during LAM processing, numerical simulation and molecular dynamics simulation methods can be used to conduct detailed research on LAM processing at the nanoscale. This method can help scholars reveal the microscopic mechanisms of material removal under the action of a laser thermal field, providing a theoretical basis for optimizing processing technology. Through these advanced research methods, laser parameters can be finely controlled to achieve precise control of the material removal process, thereby improving processing efficiency and quality. In addition to studying the machining mechanisms of LAM, more in-depth exploration is also needed on several key performance indicators of LAM technology, such as surface integrity, cutting force, and material removal rate. These indicators are directly related to the quality of processed products and the economic benefits of the processing process. Therefore, systematic research on them is crucial for promoting the practical application of LAM technology. By optimizing these key performance indicators, the competitiveness of LAM technology in industrial production can be further enhanced. Finally, in order to promote the industrial application of LAM technology, it is particularly urgent to develop laser-assisted machining systems that are easy to operate and highly integrated. This system will make LAM technology more practical, facilitate rapid deployment and use on production lines, and promote the ultra-precision machining of more difficult-to-machine materials. With the advancement of technology and the development of systems, LAM technology is expected to play a more important role in the field of industrial manufacturing, bringing revolutionary changes to precision manufacturing.

## Figures and Tables

**Figure 1 micromachines-16-00173-f001:**
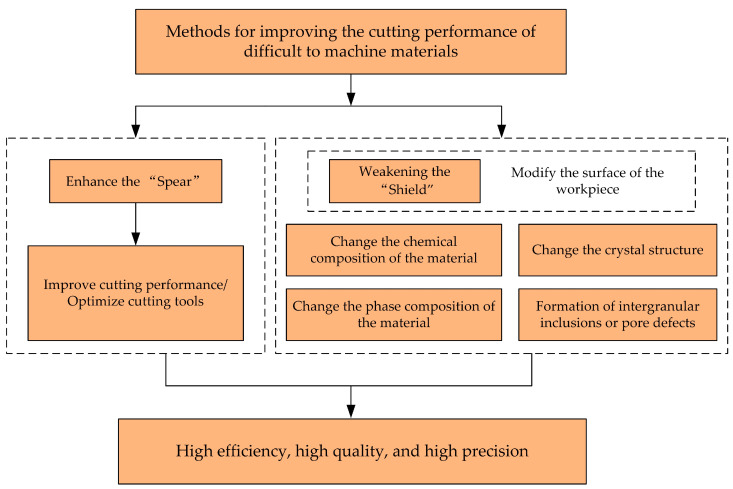
Methods for improving the machinability of difficult-to-machine materials.

**Figure 2 micromachines-16-00173-f002:**
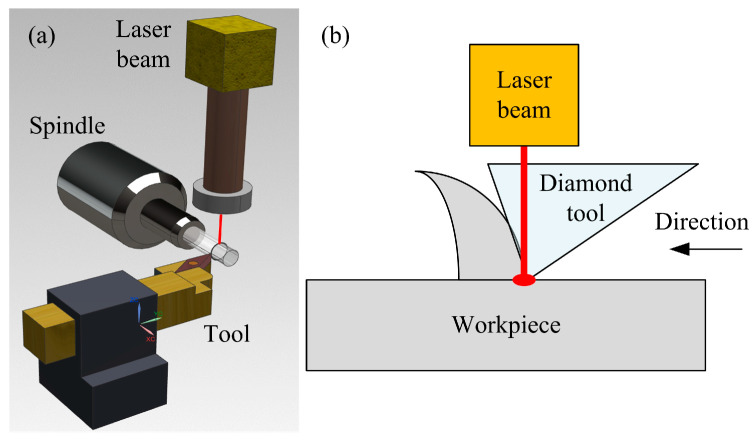
Principle of laser-assisted cutting processing. (**a**) Traditional laser-assisted cutting. (**b**) In situ laser-assisted cutting.

**Figure 3 micromachines-16-00173-f003:**
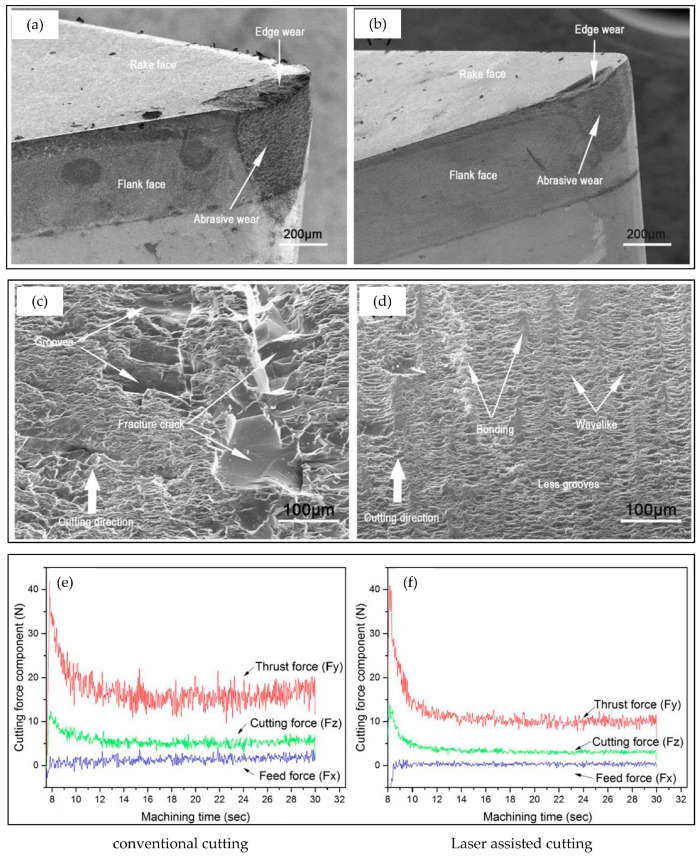
(**a**–**f**) Comparison of tool wear, machining quality, and cutting force data between conventional cutting and laser-assisted cutting [29].

**Figure 4 micromachines-16-00173-f004:**
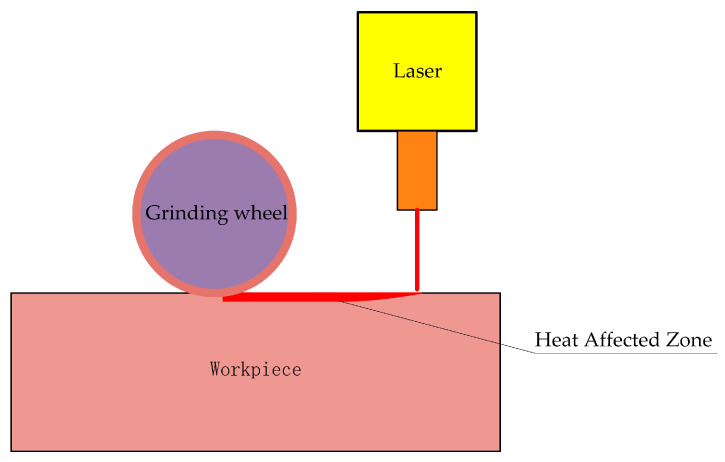
Principle of laser-assisted grinding processing.

**Figure 5 micromachines-16-00173-f005:**
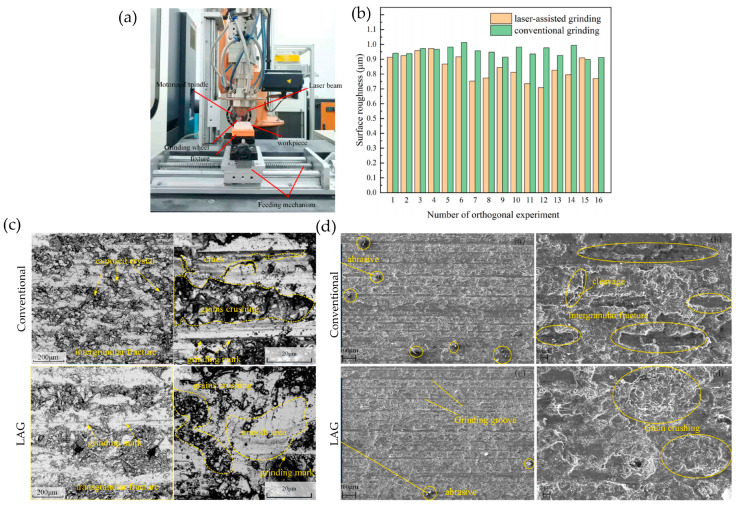
(**a**–**d**) Comparison of machining quality data between conventional grinding and laser-assisted grinding [41].

**Figure 6 micromachines-16-00173-f006:**
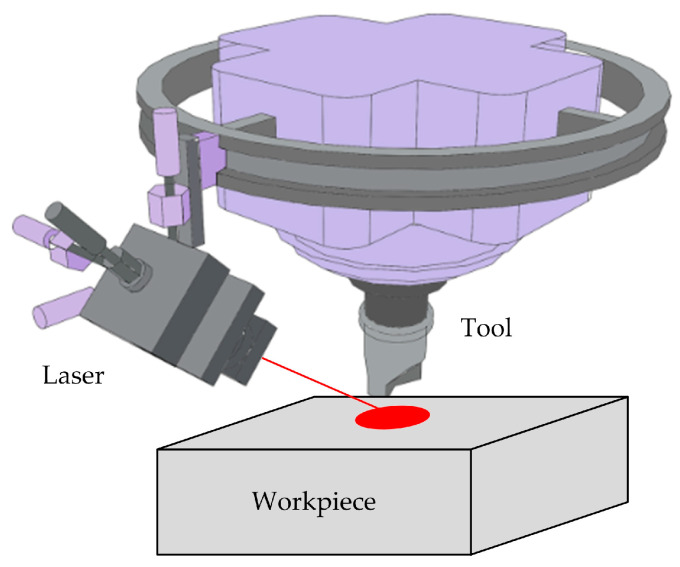
Principle of laser-assisted milling processing.

**Figure 7 micromachines-16-00173-f007:**
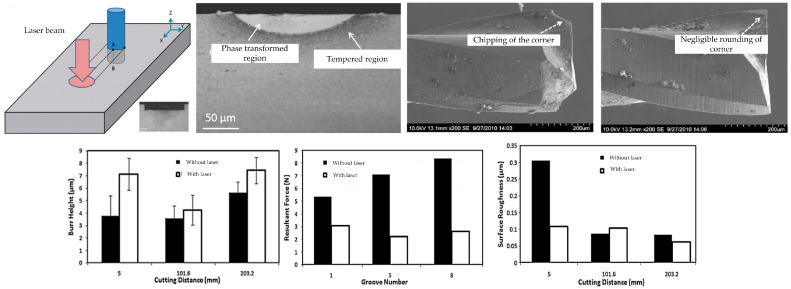
Comparison of tool wear and machining quality data between conventional milling and laser-assisted milling [56].

**Figure 8 micromachines-16-00173-f008:**
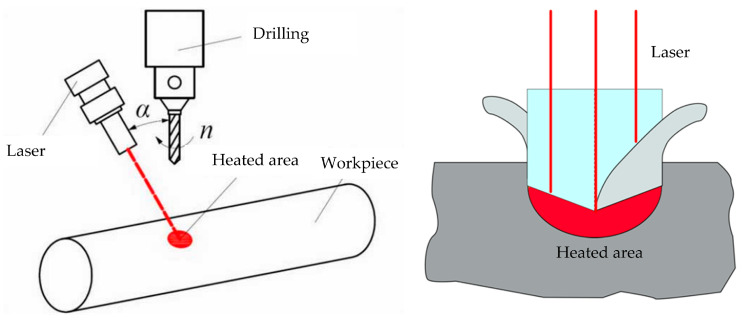
Principle of laser-assisted drilling milling processing.

**Figure 9 micromachines-16-00173-f009:**
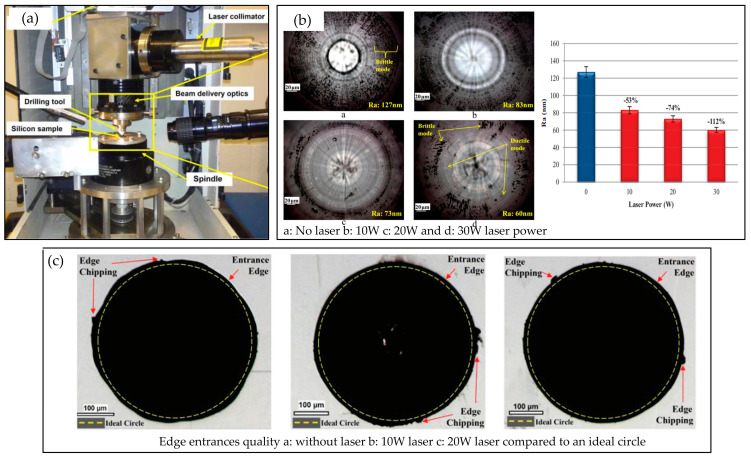
(**a**–**c**) Comparison of machining quality data between conventional drilling and laser-assisted drilling [63].

**Figure 10 micromachines-16-00173-f010:**
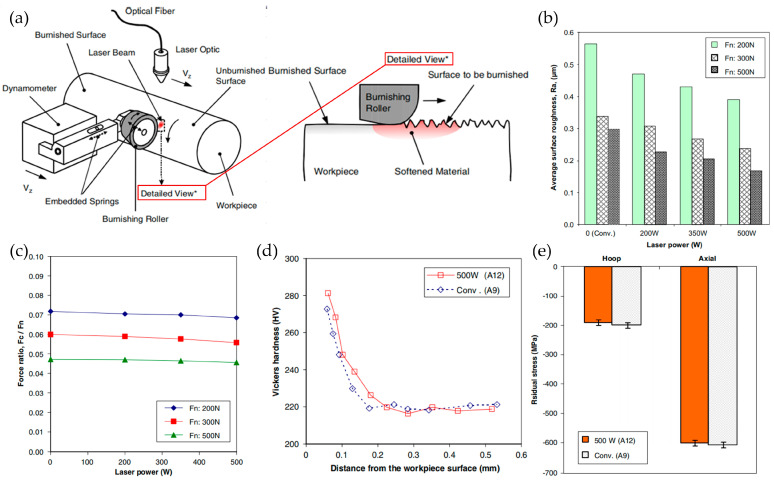
(**a**–**e**) Comparison of quality data between conventional polishing and laser-assisted polishing processing [35].

**Table 1 micromachines-16-00173-t001:** The laser specifications (such as wavelength, power, and pulse parameters) and the advantages of laser-assisted processing technology.

Parameter	Describe	Typical Value/Range
Wavelength	The electromagnetic wavelength of the laser beam determines the color and frequency of the laser.	Common wavelengths: 1064 nm (infrared), 532 nm (green light), 355 nm (UV), 1970 nm (mid-infrared), etc.
Output power	Energy output by the laser per unit time, usually in watts (W).	3000 W (industrial cutting), 600 W (stainless steel cutting), 1.5 W (femtosecond laser).
Pulse length	The duration of a single laser pulse is usually measured in nanoseconds (ns), picosecond (ps), or femtosecond (fs).	0.2–20 ms (industrial cutting), <300 fs (femtosecond laser).
Pulse repetition frequency	Repeat of laser pulses per unit time, usually in Hertz (Hz).	0.1–500 Hz (industrial cutting), 80 MHz (femtosecond laser).
High precision	The laser beam can be focused to the micron level or even nanoscale level, achieving high precision processing; the cutting seam width can be as low as 0.1 mm, and the cutting surface is smooth without burr.	
High efficiency	Laser processing has a fast speed and efficiency dozens of times higher than traditional mechanical processing, making it particularly suitable for new product development and mass production.	
Non-contact processing	Laser processing does not directly contact the workpiece, has no mechanical force, avoids tool wear and workpiece deformation, and reduces vibration and noise.	
Wide processing range	Capable of processing metals (such as stainless steel, aluminum, copper), non-metals (such as plastics, ceramics, glass), composite materials, and wood, suitable for high-hardness, high-brittleness, and high-melting-point materials.	

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
