# Peer review of "Research Progress on Laser-Assisted Precision Machining Technology"

_micromachines, 2025, doi:10.3390/mi16020173_

Round 1
Reviewer 1 Report
Comments and Suggestions for Authors
The literature review is quite complete, however, if the paper can be improved in the following areas, it would add more value to the readers:
1. The author introduces the current development status of laser-assisted processing technology in detail, but recommends supplementing the problems that still need to be further improved and overcome in the future development of LAC, LAMill and LAD technologies.
2. It is recommended to compile a general table before the conclusion to describe the laser specifications (such as wavelength, power, pulse parameters) and advantages of different laser-assisted processing technologies.
Author Response
Q1.It is recommended to compile a general table before the conclusion to describe the laser specifications (such as wavelength, power, pulse parameters) and advantages of different laser-assisted processing technologies.
A1:Add ded according to your opinion.
Laser-assisted machining technology has been widely used in manufacturing due to its advantages such as high precision, high efficiency, non-contact processing, and wide applicability. However, it also has limitations such as high equipment costs and high energy consumption, requiring the selection of appropriate technical solutions based on specific application scenarios. Table 1 describes the laser specifications (such as wavelength, power, and pulse parameters) and the advantages of laser-assisted processing technology.
Table 1. The laser specifications (such as wavelength, power, and pulse parameters) and the advantages of laser-assisted processing technology.
|
Parameter |
Describe |
Typical value / range |
|
Wavelengthï¼› |
The electromagnetic wavelength of the laser beam determines the color and frequency of the laser. |
Common wavelengths: 1064nm (infrared), 532nm (green light), 355nm (UV), 1970nm (mid-infrared), etc. |
|
Output power |
Energy output by the laser per unit time, usually in watts (W). |
3000W (industrial cutting), 600W (stainless steel cutting), 1.5W (femtosecond laser). |
|
Pulse length |
The duration of a single laser pulse is usually measured in nanoseconds (ns), picosecond (ps), or femtosecond (fs). |
0.2ms-20ms (industrial cutting), <300fs (femtosecond laser). |
|
Pulse repetition frequency |
Repeat of laser pulses per unit time, usually in Hertz (Hz). |
0.1Hz-500Hz (industrial cutting), 80 MHz (femtosecond laser). |
|
Advantage |
Specific description |
|
|
High-precision |
The laser beam can be focused to micron level or even nanoscale level, achieving high precision processing, the cutting seam width can be as low as 0.1mm, and the cutting surface is smooth without burr. |
|
|
High efficiency |
Laser processing has a fast speed and efficiency dozens of times higher than traditional mechanical processing, making it particularly suitable for new product development and mass production. |
|
|
Non contact processing |
Laser processing does not directly contact the workpiece, has no mechanical force, avoids tool wear and workpiece deformation, and reduces vibration and noise. |
|
|
Wide processing range |
Capable of processing metals (such as stainless steel, aluminum, copper), non metals (such as plastics, ceramics, glass), composite materials, and wood, suitable for high hardness, high brittleness, and high melting point materials. |
|
Reviewer 2 Report
Comments and Suggestions for Authors
In this review paper the authors report on modern technology in laser assisted machining technology. In particular, they cover laser assisted cutting, laser assisted grinding, laser assisted milling, and laser assisted drilling. Principles for each of these techniques are briefly explained with appropriate illustrations, and surveys of the most recent developments are given. The authors cited over 70 papers, mostly from the last five to ten years.
The paper is well written and gives a nice introduction to the modern laser assisted technologies.
I didn’t find significant errors and recommend the work for publication as it is.
Author Response
Q1.In this review paper the authors report on modern technology in laser assisted machining technology. In particular, they cover laser assisted cutting, laser assisted grinding, laser assisted milling, and laser assisted drilling. Principles for each of these techniques are briefly explained with appropriate illustrations, and surveys of the most recent developments are given. The authors cited over 70 papers, mostly from the last five to ten years.
A1:According to your opinion: there are 40 documents cited in the past 2019-2024.
Q2.The paper is well written and gives a nice introduction to the modern laser assisted technologies. I didn’t find significant errors and recommend the work for publication as it is.
A2:Thank you for your recognition. I will make persistent efforts.
Reviewer 3 Report
Comments and Suggestions for Authors
The manuscript is well-written, well-structured, and provides valuable insights for those working in the field of laser materials processing. However, there are areas that could benefit from further improvement:
-
Expand References: The manuscript would be significantly strengthened by including more references. There is a wealth of related work in the literature that needs to be acknowledged and addressed to provide a comprehensive context for the study.
-
Address Emerging Topics: Sub-micron laser machining is a novel and rapidly evolving topic. It would be beneficial to discuss its relevance and implications in the manuscript to provide a broader perspective and enhance its impact.
-
Improve Figure Readability: The text in some figures is difficult to read. Please ensure that all figure text is legible, with appropriate font size and clarity, to improve the overall presentation and accessibility of the data.
Author Response
A1.Address Emerging Topics: Sub-micron laser machining is a novel and rapidly evolving topic. It would be beneficial to discuss its relevance and implications in the manuscript to provide a broader perspective and enhance its impact.
Q1:Thank you for your valuable advice. Unfortunately, I have not found the valuable sub-micron laser processing literature. If allowed, I hope you can put forward more valuable suggestions.
A2.Improve Figure Readability: The text in some figures is difficult to read. Please ensure that all figure text is legible, with appropriate font size and clarity, to improve the overall presentation and accessibility of the data.
Q2:Thank you for your valuable comments: a bunch of pictures from the article were reprocessed.(Figure3, Figure5, Figure7.)
Round 2
Reviewer 3 Report
Comments and Suggestions for Authors
Although sub-micro machining is not feasible using conventional laser methods due to the wavelength limitations and heat accumulation in the heat-affected zone (HAZ), numerous studies in the literature have explored laser sub-micro machining using more advanced setups and configurations. Here are a few examples:
- Tan, B., A. Dalili, and K. Venkatakrishnan. "High repetition rate femtosecond laser nano-machining of thin films." Applied Physics A 95 (2009): 537-545.
- Krechel, Hana H. "Fluctuations in Thermal Field of Workpiece due to Multiple Pulses in Ultra-High Precision Laser Nano-Machining." (2015).
- Chung, Haseung, Katsuo Kurabayashi, and Suman Das. "Laser Nano-Machining Using Near-Field Optics." Integrated Nanosystems: Design, Synthesis, and Applications. Vol. 41774. (2004).